# Common origin of sterol biosynthesis points to a feeding strategy shift in Neoproterozoic animals

T. Brunoir[1], C. Mulligan[1], A. Sistiaga [2], K. M. Vuu [3], P. M. Shih [3,4], S. S. O'Reilly [5], R. E. Summons [6] & D. A. Gold [1] ✉

Steranes preserved in sedimentary rocks serve as molecular fossils, which are thought to record the expansion of eukaryote life through the Neoproterozoic Era ( ~ 1000-541 Ma). Scientists hypothesize that ancient $C_{27}$ steranes originated from cholesterol, the major sterol produced by living red algae and animals. Similarly, $C_{28}$ and $C_{29}$ steranes are thought to be derived from the sterols of prehistoric fungi, green algae, and other microbial eukaryotes. However, recent work on annelid worms–an advanced group of eumetazoan animals–shows that they are also capable of producing $C_{28}$ and $C_{29}$ sterols. In this paper, we explore the evolutionary history of the *24-C sterol methyltransferase (smt)* gene in animals, which is required to make $C_{28+}$ sterols. We find evidence that the *smt* gene was vertically inherited through animals, suggesting early eumetazoans were capable of $C_{28+}$ sterol synthesis. Our molecular clock of the animal *smt* gene demonstrates that its diversification coincides with the rise of $C_{28}$ and $C_{29}$ steranes in the Neoproterozoic. This study supports the hypothesis that early eumetazoans were capable of making $C_{28+}$ sterols and that many animal lineages independently abandoned its biosynthesis around the end-Neoproterozoic, coinciding with the rise of abundant eukaryotic prey.

Organic compounds preserved in rocks–known as molecular fossils or biomarkers–offer a unique window into the early evolution of life. Compared to other biological molecules, such as nucleic acids and proteins, lipids are particularly resistant to degradation, with structural features that can be preserved in the geologic record for hundreds of millions, potentially billions, of years. Despite the ever-present risks of contamination and diagenetic alteration[1–3] the biomarker field is coalescing around best practices, and clear patterns are emerging[4]. Steranes, the diagenetic remains of sterol lipids found in eukaryotic cell membranes, have proven particularly informative in Neoproterozoic-age rocks (~1000-541 Ma), where animal fossils are vexingly scarce (Fig. 1). Sterols are present in all eukaryotes and perform essential functions within the cell membrane. In eumetazoan animals and multicellular red algae, these functions are generally performed by the 27-carbon ($C_{27}$) sterol, cholesterol, while $C_{28}$ sterols are the dominant sterols in most fungi, and $C_{29}$ sterols are common in green algae and plants[5,6]. These sterols are observed widely throughout the eukaryotic tree of life, suggesting their presence in the last common ancestor[7]. In the geologic record, early proto-steranes have been identified in the Barney Creek Formation from ~1640 Ma, and can be found in rocks until ~850 Ma[8,9]. $C_{27}$ steranes–the diagenetic products of cholesterol–become abundant in Neoproterozoic-age rocks starting around 850 Ma, while $C_{28}$ and $C_{29}$ steranes become prominent in the interglacial period ~663–635 Ma[4,10]. The first appearance of steranes is hypothesized to

---

[1]Department of Earth and Planetary Sciences, University of California, Davis, Davis, CA, USA. [2]Globe Institute, University of Copenhagen, Copenhagen, Denmark. [3]Joint BioEnergy Institute, Lawrence Berkeley National Laboratory, Berkeley, CA, USA. [4]Department of Plant and Microbial Biology, University of California, Berkeley, Berkeley, CA, USA. [5]Department of Life Sciences, Atlantic Technological University, ATU Sligo, Ash Lane, Sligo, Ireland. [6]Department of Earth, Atmospheric, and Planetary Sciences. Massachusetts Institute of Technology, Cambridge, MA, USA. ✉e-mail: dgold@ucdavis.edu

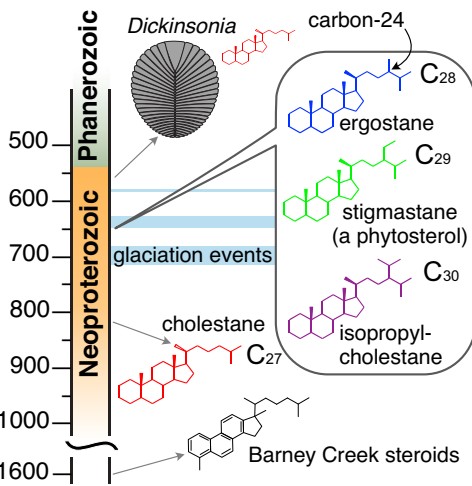

**Fig. 1 | Overview of the Neoproterozoic sterane record.** The geologic timescale is provided on the left, with dates in hundreds of millions of years. Arrows indicate the approximate time when various biomarkers become detectable above background thresholds. The site where sterols are normally methylated by the gene *24-C sterol methyltransferase* (carbon-24) is noted on ergostane with an arrow.

represent the ecological expansion of early eukaryotes (possibly red algae), with fungi and green algae expanding substantially around 663 Ma[4,10]. The $C_{30}$ sterane isopropylcholestane also occurs around this time and could represent a biomarker for sponges or rhizarian protists[11–13]. Recently, steranes and bacterial triterpenoids have also been used to taxonomically constrain enigmatic fossils from the Neoproterozoic, including the assignment of *Beltanelliformis* as a colonial cyanobacterium and *Dickinsonia* as an animal[14–16]. Taken together, these steranes paint a picture of increasing eukaryote diversity leading up to the Cambrian (~541–485 Ma) radiation of fossils.

Despite recent advances, the potential to taxonomically constrain various biomarkers is both complex and understudied. For example, there are many living groups of eukaryotes besides green algae that produce $C_{29}$ sterols whose ancestors could have been responsible for Neoproterozoic steranes–including various fungi, choanoflagellates, brown algae, and ichthyosporeans[11]. Thus the linking of $C_{29}$ steranes in the rock record to prehistoric green algae would be premature. The same can be said of all major steranes in the Neoproterozoic. Comparative genomics can help resolve this uncertainty, as it offers a powerful tool for identifying organisms capable of synthesizing various lipids, and determining when in Earth's history they evolved such abilities[17]. In eukaryotes, the gene *24-C sterol methyltransferase* (*smt*) encodes an enzyme that adds methyl groups to carbon-24 of the sterol side chain (signified as C-24; see Fig. 1) and is responsible for the production of $C_{28+}$ sterols. With increased genomic sampling the number of candidate *smt* genes has been expanding. This study was initiated based on a putative *smt* from the annelid *Capitella teleta*[2,18]. Unlike sea sponges, annelid worms are part of the Eumetazoa, a clade of animals whose cell membranes are generally dominated by cholesterol, and who are thought to have lost the ability to make $C_{28+}$ sterols[7,18]. Since we began the study of *smt* in *C. Teleta*, another study by Michellod et al. has demonstrated the synthesis of $C_{28/29}$ sterols in a different annelid, *Olavius algarvensis*, and reported *smt* genes in many animals[19]. The goal of this study is to determine the veracity of the *C. teleta smt* and reconstruct its evolutionary history. If *smt* genes have been conserved in annelid worms from the earliest animals, then the history of $C_{28+}$ sterol biosynthesis in animals may be far more complex than currently appreciated.

In this work, we show that *smt* genes responsible for $C_{28+}$ sterol biosynthesis are present in diverse annelid worms, as well as several other eumetazoan clades. Using yeast gene rescue experiments, we demonstrate that the *C. teleta* SMT protein is capable of synthesizing $C_{28}$ sterols, providing evidence that SMT function has been conserved through annelid evolution. Careful vetting of the animal *smt* tree supports vertical inheritance of the gene from a common ancestor, as opposed to inheritance via horizontal gene transfer[19]. Finally, we produce molecular clocks to demonstrate that a vertically inherited SMT was spreading through eumetazoans during the Neoproterozoic, with most lineages losing the gene around this time. Our work suggests that the first eumetazoans were capable of $C_{28+}$ sterol biosynthesis and that many lineages independently abandoned the biosynthesis of complex sterols concurrent with the rise of their sterol-rich prey, as evidenced by a rise in algal biomarkers during this time.

## Results

### Sterol methyltransferases are found across the annelids

Annelids are an ecologically diverse phylum containing ~20,000 described species, yet they are severely undersampled in genetic sequence databases such as the National Center for Biotechnology Information's (NCBI) Genbank. To determine how prevalent *smt* genes are in annelids, we queried a number of unannotated genomes and transcriptomes for candidate genes (see Methods). We ultimately recovered 27 *smt* homologs from 20 species of annelids. These putative genes are found across major clades, and their translated proteins contain the conserved domains expected in functional sterol methyltransferases (Fig. 2). In many cases, we discovered multiple *smt* genes in the same species, indicating multiple rounds of gene duplications within the group. Annelid SMT proteins form a well-supported clade in our tree-building analysis, suggesting they are derived from a common ancestor (Supplementary Figs. 1–3). This demonstrates that sterol methyltransferases are indeed present across diverse annelid worms.

We used I-TASSER[20] to model several annelid SMT proteins, comparing their tertiary structure to each other and to better-studied SMTs from yeast and plants (Fig. 3). In particular, we analyzed the SMT from *C. teleta* (Fig. 3A), as well as two SMTs recovered from the model annelid *Platynereis dumerilii* (Fig. 3B, C). These were compared to an SMT from the fungus *Saccharomyces cerevisiae* (also known as ERG6), which is required for the synthesis of the $C_{28}$ sterol ergosterol (Fig. 3D), and the SMT2 protein in the plant *Arabidopsis thaliana*, a bifunctional enzyme that can generate $C_{28}$ and $C_{29}$ sterols (Fig. 3H)[21]. Root-mean-square deviation (RMSD) of atomic positions suggests that annelid SMTs are highly similar to both ERG6 and SMT2, and in fact, are more similar to both proteins than ERG6 and SMT2 are to each other (RMSD = 2.412). The methyltransferase domains–the region where sterol binding occurs[22]–demonstrate high conservation across proteins, while the C-terminal domain shows greater variability (3E-G,I-K). The *C. teleta* SMT and one of the two *P. dumerilii* proteins appear more structurally similar to the bifunctional SMT2 than ERG6, as suggested by their lower RMSD scores, and is particularly noticeable in the alpha helices in the C-terminal domain. This supports recent results by Michellod et al. demonstrating that SMTs of the annelids *Olavius* and *Inanidrilus* can bifunctionally generate $C_{28}$ and $C_{29}$ sterols[19], and suggests this ability is commonplace across annelids.

### Functional analysis of annelid SMTs demonstrates their ability to methylate sterols

After modeling the *Capitella teleta* SMT, we assessed whether or not it is capable of methylating sterols at the C-24 position. To test this we performed a rescue experiment, introducing the *C. teleta smt* into a strain of *S. cerevisiae* yeast lacking the wild-type gene (commonly known as ERG6). ERG6⁻ yeasts are viable but cannot produce ergosterol and have a slower rate of growth than their wild-type counterparts[23]. We found that ERG6⁻ *S. cerevisiae* is capable of producing ergosterol with the addition of the *C. teleta* gene (Fig. 4). This functional analysis complements Michellod et al., demonstrating that multiple annelid species can methylate sterols[19].

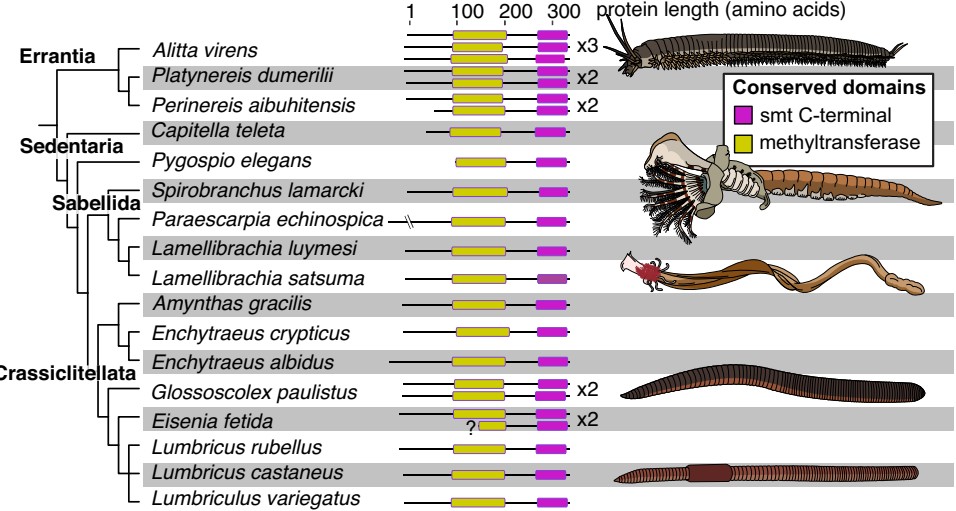

**Fig. 2 | Distribution and structure of annelid SMT proteins.** This figure is restricted to annelid SMT proteins containing both conserved domains (i.e. excluding partial sequences). (Left) Phylogeny of annelids in our study with recovered full-length SMT proteins. (Right) Image displaying the number and size of SMT proteins. The position and length of conserved domains is visualized with colored boxes. Source data are provided as a Source Data file.

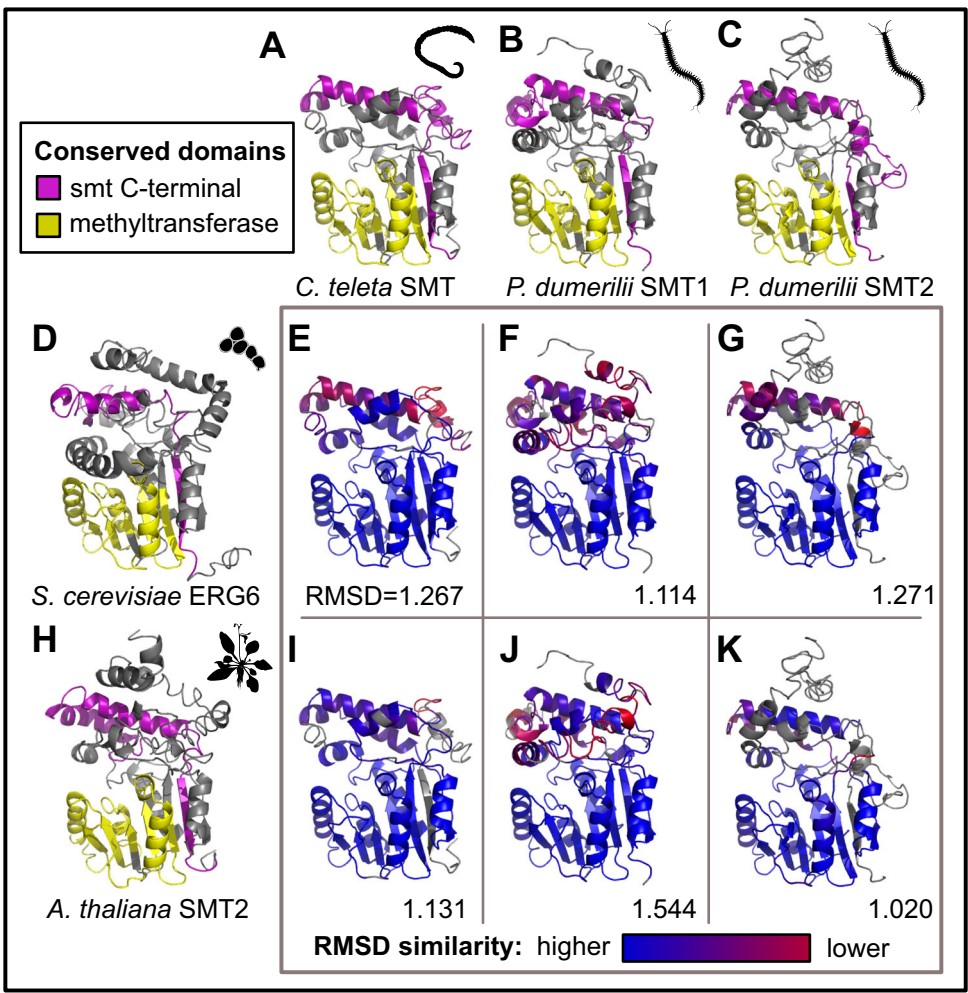

**Fig. 3 | Modeling of select annelid SMT proteins. A–C** best-scoring models for the SMT protein in *C. teleta and* two SMTs in *P. dumerilli*, with conserved domains highlighted. **D** Model of *S. cerevisiae* ERG6. **E–G** Alignment of the three above annelid proteins against ERG6, colored by RMSD values. The overall RMSD score for each comparison is shown in the lower right corner. **H** Model of *A. thaliana* SMT2 protein. **I–K** Alignment of the three above annelid proteins against *A. thaliana* SMT2, colored by their RMSD values. The overall RMSD score for each comparison is shown in the lower right corner. Protein models were generated using the I-TASSER server. Source data are provided as a Source Data file.

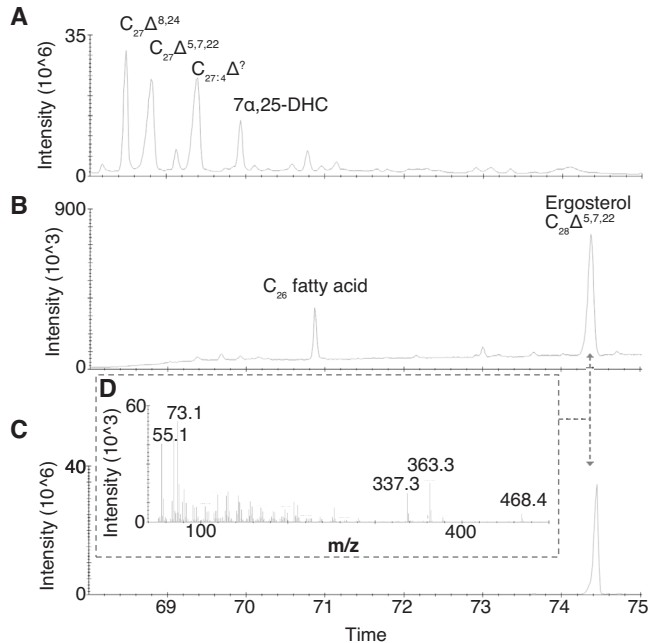

**Fig. 4 | Production of ergosterol is rescued in yeast when ERG6 is replaced by its annelid homolog. A** Partial total gas chromatogram from an ERG6 knockout yeast without transformation of the *C. teleta smt* gene. **B** Partial total ion chromatogram from an ERG6 knockout yeast with transformation of *C. teleta smt*. **C** An ergosterol standard mix. **D** The ergosterol mass spectrum from comparable peaks in (**B**) and (**C**), with characteristic fragment ions at *m/z* 468, 363, 337 for trimethylsilyl-derived samples. Source data are provided as a Source Data file.

## Phylogenetic analysis of animal SMTs supports vertical inheritance

In addition to annelids, we recovered SMT proteins from several other animal clades. Many of our SMTs came from shotgun transcript sequencing projects, which can easily be contaminated by an animal's diet and symbionts. To minimize likely contaminants in our final dataset, we made an initial phylogenetic tree and rejected any animal sequences that did not form a clade with one or more species from the same phylum (see Methods; Supplementary Figure 1). While this methodology may have inadvertently pruned real instances of horizontal gene transfer from our dataset, it also removed many genes we were able to independently confirm were contaminants. For example, some of the rejected SMTs come from animals that have publicly available genomes. In each case we failed to find the putative *smt* transcript in the relevant genome (Figure S1). Our analysis of the data in Michellod et al. suggests contamination was a problem in their phylogenetic tree as well. The molluscs provide a good case study: Michellod et al. report 13 mollusc *smt* transcripts, which evolved through a minimum of 11 separate horizontal gene transfer events (see Fig S26 from their paper). Three of the molluscs in their tree have genomes on NCBI (*Biomphalaria pfeifferi*, *Crassostrea hongkongensis*, and *Pecten maximus*); none of their genomes contains evidence of an *smt* (see Methods). An additional two species do not have genome assemblies on NCBI (*Haliotis tuberculata* and *Elysia cornigera*), but other members of the same genus do. Again, we found no evidence for *smt* genes in these genomes. This strongly suggests that many, perhaps all, of the putative mollusc *smt* transcripts are contaminants from shotgun sequencing projects. No mollusc *smt* passed vetting in our analysis. It is possible that future work will demonstrate that some of the *smt* genes we rejected represent true animal sequences, but false positives caused by contamination are clearly a major issue. Our results challenge the hypothesis that rampant horizontal gene transfer explains the presence of *smt* genes in annelids and other eumetazoans[19].

Following our vetting process we were left with *smt* genes from three clades of eumetazoans–annelids, rotifers and stony corals. Nematode worms also have an SMT-like protein[18], but we did not include it in our analysis because the gene did not cluster with other eukaryote SMTs in our phylogeny, and because the protein is known to catalyze a C-4 methylation step that is distinct from the C-24 methylation seen in genuine SMTs[24]. Rotifers are a group of near-microscopic animals that are distant relatives to annelids. A recent project assembling 31 rotifer genomes resulted in a large number of *smt* genes from the genera *Adineta*, *Rotaria*, and *Didymodactylos*[25]. Regarding stony corals (clade Scleractinia), one SMT-like protein has been annotated in the *Orbicella faveolata* genome (NCBI Accession: XP_020604180.1), but it lacks the C-terminal domain found in functional SMTs. However, we recovered multiple SMTs from other coral transcriptomes that retain both domains. We were able to map one of these genes (NCBI Accession: FX438716.1) to the genome of the coral *Porites australiensis* (Supplementary Figure 4). A second candidate SMT from *P. australiensis* did not map to the genome, and shows high sequence similarity to the coral symbiont *Symbiodinium*, again demonstrating that our methods can distinguish between authentic and contaminating sequences.

After removing questionable sequences from our dataset, all animal SMT proteins, except rotifers, formed a monophyletic clade (Supplementary Fig. 2). While rotifers did not cluster with the rest of the animals, the nodes separating these two groups are poorly supported. Following gene tree / species tree reconciliation (a process where nodes with low statistical support in a gene tree are rearranged to parsimoniously reflect the known species tree), all animal SMTs formed a single clade (Supplementary Figure 3). We therefore find no compelling support for horizontal gene transfer in animal SMTs, where we would expect to find a clade of animal SMTs that clusters with non-animals with strong statistical support.

Notably, the animal SMT tree does not replicate the species tree exactly. Instead, our final tree (Figure S3) prefers two clades: one containing sea sponges and corals, and another containing sea sponges, rotifers, and annelids. Our result is driven by strong statistical support for a clade including Heterosclermorph sponges and cnidarians to the exclusion of other sponges. It is possible that better taxon sampling of sponges will ultimately demonstrate that this duplication inference is driven by an error in the gene tree. Alternatively, this could suggest that the sponge gene duplication that was first hypothesized in Gold et al. (2016) is actually more ancient than anticipated, predating the divergence of living animals. Given the data currently available, we conclude that the best interpretation of animal SMTs is vertical inheritance from a common ancestor, with a gene duplication event occurring before sea sponges diverged from the other living animals.

## A molecular clock suggests eumetazoan SMTs diversified in the Neoproterozoic

To test whether the diversification of eumetazoan SMTs coincides with the Neoproterozoic biomarker record, we generated a gene-centered molecular clock using 107 vetted SMT sequences and nine fossil calibrations (see Supplementary Methods for details and justification of calibrations). The results are summarized in Fig. 5, with a more detailed output provided in Supplementary Figure 5. As this clock is based on a single gene, the error bars are large and the exact dates should be interpreted with caution. However, our results are consistent with multigene, species-level molecular clocks, which place the origin of eukaryotes between ~1800 and 1600 Ma and animals between ~828 and 572 Ma[26–28]. As mentioned earlier, many annelids and rotifers have multiple SMTs. Our tree suggests this is primarily due to lineage-specific gene duplication events (marked by circles in Fig. 5), which have occurred throughout the Phanerozoic (<541 Ma). Our results demonstrate that the two eumetazoan *smt* genes overlap with the appearance of $C_{28}$ and $C_{29}$ steranes in the fossil record ~663-635 Ma. We re-ran the molecular clock using an alternate topology where all

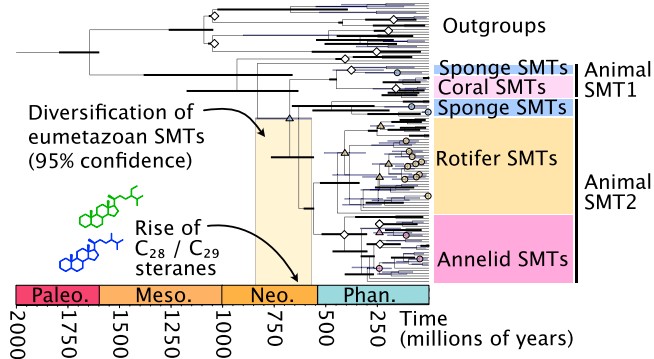

**Fig. 5 | The molecular clock of animal SMT proteins.** This SMT-centered molecular clock was generated using BEAST. White diamonds represent fossil calibrations, which are described in detail in the Supplementary Methods. Putative gene duplication events in the animals have been labeled at relevant nodes with colored shapes: circles represent gene duplication events that are specific to one species in our study; triangles represent events that include multiple species. White diamonds represent fossil calibrations. Phan. = Phanerozoic; Neo. = Neoproterozoic; Meso. = Mesoproterozoic; Paleo. = Paleoproterozoic. Source data are provided as a Source Data file.

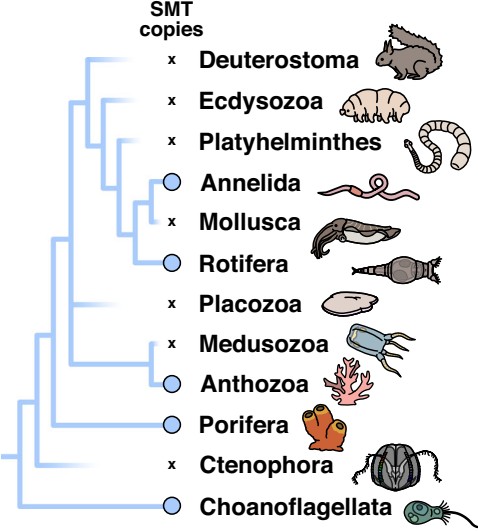

**Fig. 6 | Summary of *smt* gene loss in the animals.** A simplified animal phylogeny, with the hypothesized presence of *smt* visualized with blue lines. This tree provides a conservative estimate of the number of *smt* losses, as it excludes many understudied animal phyla that may also lack the protein.

sponge sequences are monophyletic (i.e. no ancestral gene duplication). The results of this alternate topology still suggest the diversification of eumetazoan SMTs coincides with the rise of $C_{28/29}$ steranes (Supplementary Fig. 6). We can therefore conclude that eumetazoans had functional SMTs concurrent with, and likely prior to, the rise of complex steranes in the Neoproterozoic.

## Discussion

Our research suggests that the *smt* gene necessary to synthesize complex sterols existed in the ancestor of Eumetazoa, and was retained in some lineages long after their diversification in the Cambrian. These results contradict the hypothesis that *smt* genes evolved in eumetazoans as a result of horizontal gene transfer[19]. If we are correct, then the observation that most living eumetazoans lack *smt* genes is a function of extensive gene loss[18]. The retention of *smt* genes in annelids via vertical inheritance necessitates its loss in at least seven major animal groups (Fig. 6). The actual number of losses is likely an order of magnitude higher, given the large number of coral and annelid genomes that lack SMT proteins, as well as the possible loss of SMTs in the minor animal phyla, which are poorly represented in genetic sequence databases. The presence of one or more *smt* genes in the ancestor of Eumetazoa raises important questions about the sterols biosynthesized by the earliest animals. What lipid(s) did the first animal SMT—or, if our scenario is correct, the first pair of SMTs—synthesize? Was it bifunctional like those seen in some sea sponges and annelids? Answers to these questions will help determine whether early eumetazoans could biosynthesize some of the exotic steranes found in sedimentary rocks.

Additionally, why the *smt* gene has been retained and independently duplicated in many eumetazoan lineages remains an open question. For rotifers, the large number of *smt* duplicates may lay in part with their unusual reproductive strategy; the species in our dataset reproduce asexually and exhibit tetraploidy[29]. Consequently, their genomes are more like plants than other animals, and like plants, duplicate SMT proteins are likely to be partially redundant[30]. The presence of multiple SMTs in some annelids is harder to explain. For example, two SMT copies have been retained in hesionoid worms (*Alitta virens*; *Perinereis aibuhitensis*, and; *Platynereis dumerilii*) for over 300 million years, and must therefore confer some function. There is little work on the lipids of annelids, and fewer where the environment and diet is well constrained. Most annelids studied have cholesterol as their

dominant sterol, but many contain complex $C_{28+}$ sterols as well[31–34]. In *C. teleta*, the abundance of sterols can vary dramatically depending on diet and life stage, in some cases demonstrating higher percentages of $C_{28+}$ sterols in their bodies than exists in their food[35]. This variation suggests a strong biological control on endogenous sterol uptake. Therefore, annelid SMTs may play specific roles based on the animal's development and/or environment. While there is a general relationship between SMT copy number and the type of sterols synthesized, we know of enough lineages with bifunctional enzymes to prevent any a priori predictions about the sterols produced by these animals[19,30,36,37]. The selective advantage of retaining SMTs in some eumetazoan lineages, when the vast majority have abandoned the protein, will require further research on their use and function in specific species.

Regardless of the reason why *smt* genes were retained and expanded in some animals, our results suggest that the biosynthesis of complex sterols is an ancestral and ancient trait. This work does not discount fossil evidence of $C_{27}$-dominated early animals, such as dickinsoniamorphs, but when $C_{28+}$ steranes are found in eumetaozoan fossils, the possibility these organisms were synthesizing higher sterols should be considered[15,16]. In addition, our conclusions have distinct implications for the evolution of animal sterols and feeding strategies. In our scenario of vertical inheritance, multiple animal groups lost the ability to synthesize higher sterols across the Neoproterozoic Cambrian transition, which we hypothesize was caused by environmental changes. With the radiation of microbial eukaryotes after the Sturtian glaciation—as documented by the rise in $C_{28/29}$ steranes—early animals had a novel and abundant source for sterols[10]. The Neoproterozoic was also a time of fluctuating oxygen levels, and it has been demonstrated that some eukaryotes will switch to exogenous sterols under anaerobic conditions[38]. As ocean chemistry shifted and feeding strategies diversified, many Precambrian animal groups independently abandoned sterol modification (or sterol biosynthesis altogether). In the alternative scenario where annelids, rotifers, and cnidarians each received their *smt* genes through horizontal gene transfer, our molecular clock suggests these events occurred in the Phanerozoic (Fig. 5). The presence of *smt* genes in these animals would therefore have no relevance to our interpretation of the biomarker record or Precambrian evolution. Adjudicating between these two competing hypotheses is therefore critical to interpreting the significance of eumetazoan SMTs. If animal SMTs have been vertically inherited, then

understanding their evolutionary history and usage will reveal insights into the origins of animal feeding strategies.

## Methods

This research complies with all relevant ethical regulations of the University of California.

### Collection of putative animal SMTs

The ERG6 protein from *Saccharomyces cerevisiae* (NCBI accession: P25087.4) was used as a query for all database searches. We queried the NCBI Transcriptome Shotgun Archive (TSA) using tBLASTn, restricting our analysis to animals (taxid:33208). A similar search was performed on the non-redundant protein database using BLASTp. Transcripts from the TSA search were converted into proteins using the TransDecoder (v5.5.0) program packaged with Trinity[39]. Sequences were then separated by species, and redundant proteins were removed using CD-HIT (v4.8.1) with a 90% similarity cutoff[40]. Conserved domains were identified in the remaining proteins using the pfam_scan Perl script included in BioConda, which uses HMMER (v3.4) to compare proteins against the PFAM-A database[41,42]. Methyltransferase and C-terminal domains were extracted independently using SAMTOOLS (v1.13) and aligned with MAAFT (v7.490); they were combined again with FASconCAT-G and cleaned with trimAl (v1.4)[43,44]. These aligned sequences were used for further annotation and tree building. Relevant files and code can be found in folder 2 on GitHub.

### Vetting of SMTs

Contamination was a major concern with data downloaded from the NCBI TSA database, so a thorough vetting process was used to find real SMT sequences for the analysis. Additional data was appended to sequence IDs to help interpret the data, including (1) NCBI's TaxIdentifier was used to add taxonomic information in the dataset, (2) top reciprocal BLASTp hit when the proteins were compared against the Uniprot Swissprot dataset, which was augmented with additional SMT sequences (XP_003387525.1, XP_004346937.1, ELU07827.1, CAF1633859.1). A selection of SMT and methyltransferase outgroups were chosen from across the eukaryotes and added to our SMT protein alignment. An initial phylogenetic tree was generated from the alignment using FastTree[45]. We then went carefully through the tree to determine which genes were likely to be genuine and which were likely to be contaminants (results illustrated and annotated in Supplementary Fig. 1). When possible, we used BLASTn to query putative animal transcripts against their respective genomes. Transcripts that were not identifiable in genomes are annotated in Supplementary Figure 1, and were treated as contaminants. Sequences were only considered if two or more genera from the same phylum formed a monophyletic clade. Once clades were defined, individual sequences were removed from a clade for the following reasons: (1) the sequence was redundant, meaning it was highly similar to another sequence from the same species, or (2) the sequence came from a species in a different phylum than the clade, making it a probable contaminant. Following this process, animal SMT-like genes were restricted to the Rotifera, Annelida, Nematoda, Porifera and Cnidaria. These sequences were extracted from the original alignment, the outgroups were added back in, and a final tree was generated using IQTree (v1.6.12)[46], which is visualized in Supplementary Figure 2. In this tree, the nematode sequences fell outside of the SMT clade, but all other putative animal SMTs remained. This clade of SMTs (highlighted in Supplementary Figure 2) Was used for downstream species tree reconciliation and molecular clock analysis. Relevant files and code can be found in folder 2 on GitHub.

### Vetting of putative mollusc sequences from Michellod et al.

We used the tBLASTn algorithm on NCBI, using yeast ERG6 (accession: NP_013706.1) as a query. The following genomes were searched as databases: *Crassostrea hongkongensis* (GCA_015776775.1), *Biomphalaria pfeifferi* (GCA_030265305.1), *Pecten maximus* (GCF_902652985.1), *Elysia chlorotica* (GCA_003991915.1), *Elysia marginata* (GCA_019649035.1), *Haliotis laevigata* (GCA_008038995.1), *Haliotis cracherodii* (GCA_022045235.1), *Haliotis rufescens* (GCA_023055435.1), and *Haliotis rubra* (GCA_003918875.1). No significant matches were returned in any of these analyses. To verify that we could recover SMTs from genomes using tBLASTn, we also queried *Monosiga brevicollis* (GCA_000002865.1), *Amphimedon queenslandica* (GCA_000090795.2), and *Capitella teleta* (GCA_000328365.1). In each of these genomes, we recovered a matching genome contig. BLASTp was also used on the *Pecten maximus* genome assembly since the genome was sufficiently annotated to perform this additional analysis. The top hit in this search was a phosphoethanolamine N-methyltransferase-like protein, again suggesting no sterol methyltransferase proteins are present in this species.

### Species tree reconciliation and molecular clock analysis

The species included in our SMT clade was downloaded from the NCBI Taxonomy Browser website. The tree was manually edited using Mesquite (v3.6.1)[47] to reflect updated taxonomic affinities based on references[48–50]. The species tree and gene were passed to NOTUNG (v2.1.5), which rearranged poorly supported branches in the gene tree to provide the most parsimonious reconciliation with the species tree[51]. A molecular clock was generated from the reconciled tree using BEAST (v1.10.5)[52]. An LG amino acid substitution model was used with a 4-site gamma heterogeneity model. An uncorrelated relaxed clock with a lognormal distribution was chosen for the clock model. The tree prior followed the speciation: yule process. Fossil calibrations were modeled as lognormal priors, and are detailed in the Supplementary Methods. BEAST was run for 10,000,000 generations, with sampling every 1,000 states. A consensus tree was generated from the results using a 25% burnin and median node heights. We then re-ran the molecular clock analysis using a different starting tree. In this second analysis sponges are monophyletic, meaning there is no SMT duplication event at the origin of animals. All input files, including the XML used for the BEAST runs, are provided in folder 5 on GitHub.

### Protein modeling

Protein structure and function predictions were performed on the I-TASSER server[20]. Sequences submitted include the *Capitella teleta* SMT protein (accession: ELU07827.1), two SMTs translated from *Platynereis dumerilii* transcripts (accessions: HALR01229039.1; HALR01261698.1), ERG6 from *Saccharomyces cerevisiae* (accession: P25087.4), and SMT2 from *Arabidopsis thaliana* (Accession: NP_173458.1). The resulting Protein Data Bank models were visualized in open-source Pymol (v2.5.0). The results from I-TASSER and the Pymol code used to generate the figures are provided in folder 3 on GitHub.

### Functional analysis

The *C. teleta smt* mRNA was codon optimized for yeast and integrated into a pPMS090 plasmid with chloramphenicol resistance and p15a origin of replication. The cassette incorporating the gene was flanked by 5' and 3' homology arms at the yeast ERG6 location and contained a LEU2 auxotroph marker. Target knockout of wild-type ERG6 in *S. cerevisiae* strain BY4742 (MATα his3Δ1 leu2Δ0 lys2Δ0 ura3Δ0) was achieved by restriction digest of the *C. teleta smt* and transforming the linearized DNA using Frozen-EZ Yeast Transformation II Kit (Zymo Research, Irvine, CA). Yeast colonies were selected on SC-LEU agar plates and grown at 30 °C for ~1-2 weeks. The presence of the *C. teleta smt* at the ERG6 location was screened by yeast colony PCR and subsequently grown in SC-LEU liquid media at 30 °C. Relevant data, including the plasmid design, are provided in folder 4 on GitHub.

## Lipid extraction

The complemented yeast cultures were extracted twice with dichloromethane (DCM)/methanol (MeOH) (9:1 v/v). After the addition of water, the DCM layers were combined, washed several times with $H_2O$, and dried over $Na_2SO_4$. The lipid extracts were then concentrated and transferred to a vial for derivatization (silylation) by reaction with N, O-Bistrifluoroacetamide (BSTFA) + 1% trimethylchlorosilane in pyridine (2 h at 70 °C). All glassware, aluminum foil, silica, quartz wool, and quartz sand were combusted at 500 °C for at least 12 hours to remove organic contamination, whereas metal tools were rinsed in MeOH and DCM.

## GC-MS analysis of sterols

1 μL of silylated samples were analyzed by gas chromatography-mass spectrometry on an Agilent 5890 GC hyphenated to an Agilent 5975 C Mass Selective Detector. The GC inlet was operated in splitless mode and fitted with a J&W DB5-MS 60 m capillary column (0.25 mm inner diameter, 250 μm film thickness). The GC temperature program was 80 °C for 2 min, ramp at 3.5 °C min-1 to 315 °C, and a final hold time of 31 min. The mass spectrometer was operated in electron impact ionization mode (70 eV), with a mass scan range from m/z 50 to 700. All solvents used were high-purity (OmniSolv), all aqueous solutions were cleaned with dichloromethane prior to use, and procedural blanks were run to monitor background contamination. MSD data is provided in folder 4 on GitHub. The two ergosterol spectra from Fig. 4B and C are illustrated in Supplementary Figure 7 for comparative purposes.

## Mapping of putative SMTs to coral genomes

Putative *smt* transcripts from the corals *Acropora* and *Porites* were downloaded along with their respective genomes from NCBI. Transcripts were mapped to the genome using GMAP (v2019-09-12)[53]. The only sequence to map to a genome was a single *smt* (Accession: FX438716.1) in *P. australiensis*. To produce a gene model, the reverse complement was mapped to the genome. We then generated a genome-based coding region annotation file with TransDecoder[39]. The input files and results of this analysis are available in folder S3 on GitHub.

## Reporting summary

Further information on research design is available in the Nature Portfolio Reporting Summary linked to this article.

## Data availability

All data used in this study, including scripts executed, input files, intermediate files, and output, are provided on GitHub at https://github.com/DavidGoldLab/2022_Annelid_SMTs. The accession numbers for all previously published genetic data can be found on GitHub in Supplementary Data file 1. The data generated in this study have been deposited in the Zenodo database under accession code https://doi.org/10.5281/zenodo.10063989[54]. Source data are provided with this paper.

## Code availability

All code used is provided on GitHub at https://github.com/DavidGoldLab/2022_Annelid_SMTs. The data generated in this study have been deposited in the Zenodo database under accession code https://doi.org/10.5281/zenodo.10063989[54].

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

## Acknowledgements

D.A.G. was supported by the National Science Foundation grant (grant no. 2044871). R.E.S. was supported by the Simons Collaboration on the Origin of Life (grant no. 290361FY18). S.S.O.R. was supported by the Irish Research Council (grant no. ELEVATEPD/2014/47) and in part by a grant from Science Foundation Ireland (grant no. 21/FFP-A/9153).

## Author contributions

D.A.G. and R.E.S. conceived the project. T.B., D.A.G., and C.M. analyzed computational genetic data. P.M.S. and K.M.V. performed yeast genetics experiments. A.S., S.S.O.R., and R.E.S. performed lipid extractions and GC-MS analysis. D.A.G. drafted the original manuscript with help from T.B. and C.M. All authors read and approved the manuscript before submission.

## Competing interests

The authors declare no competing interests.
