## [Peer Review File · Nature Communications]

Common origin of sterol biosynthesis points to a feeding strategy shift in Neoproterozoic animalsReviewers' Comments:

Reviewer #1:

Remarks to the Author:

Brunoir et al address a key issue in paleontology: how to associate molecular fossil biomarkers to specific phylogenetic groups. This is especially critical for tracking first eukaryotes and particularly first animals along the Neoproterozoic-age (~1,000-541 Ma), where animal fossils are extremely scarce. By means of phylogenomics and molecular clock studies with a gene encoding a sterol methyltransferase (smt), they suggest a gene duplication event in the earliest animals that overlaps with a prominence of C28 and C29 steranes in the fossil record ~663-635 Ma (coincident with the onset of Ediacaran period), so far attributed to fungi (C28) and green algae (C29). The results bring the authors to reject the so far accepted hypothesis that cholesterol (C27) biosynthesis is the ancestral condition of the Eumetazoa, and suggest some animal lineages as potential relevant source of C28 and C29 steranes.

This reviewer finds this work highly relevant for the study of ancient life and considers it sheds light to associate lipid biomarkers to their biological producers. Also, the work shows how phylogenomics and biogeochemistry complement each other to decipher the record of life on Earth.

Minor comments:

Introduction

The work by Michellod et al. (ref. 22) should be mentioned here, to let the reader have more data about the smt gene and its presence in annelids. (Note: this reviewer has nothing to do with that work nor the group.)

Discussion

Where the authors say:

"We therefore reject the hypothesis that cholesterol biosynthesis is the ancestral condition of the Eumetazoa", this reviewer misses additional comments on cholesterol in the new scenario depicted.

Fig. 3 legend: where it says (E-F), it should be (E-G); and where it says (I-J), it should say (I-K).

Fig. 4 legend: where it says: (A) Partial total ion chromatogram, should not say (A) Partial total gas chromatogram?

Reviewer #3:

Remarks to the Author:

In their manuscript: An ancient gene duplication in animals rewrites the molecular fossil record, Brunoir and colleagues report the discovery of the gene 24-C sterol methyltransferase (smt) required for phytosterol production in certain animals (segmented worms) and use a molecular clock estimation to conclude that the animal-specific smt derived from a Neoproterozoic gene duplication event that overlaps with a rise in the abundance of C28 and C29 steranes observed in the rock record. They use these inferences to conclude that C27 sterol-production should not be viewed as a shared trait of Neoproterozoic eumetazoans, and sterane with 28 or more carbon atoms are not necessarily derived from fungi or green algae. While admitting that these results do not necessarily contradict the emerging picture of Neoproterozoic life informed by molecular fossils, Brunoir and colleagues conclude that their findings refute the underlying hypothesis that drives the interpretive paradigm, going as far as prominently claiming a rewriting of the molecular fossil record in their manuscript title.

Considering these potentially very far-reaching inferences, I was asked to assess the implications of the new discovery for interpretations of the molecular fossil record, and will not assess the underlying genetic analyses in detail. When considering the potential relevance of the study of Brunoir and colleagues and the suitability for publication in Nature Communications, it should be noted that a very

similar study was recently published in Science (Michellod et al. 2023) and that the implications of both the Michellod et al. (2023) and the yet unpublished preprint of the here discussed study for interpretations of the molecular fossil record were briefly discussed in the accompanying opinion piece of Brocks and Bobrovskiy (2023). This means, that on the one hand the main discovery, the capacity of phytosterol-production in animals, has already been published (and exceptions to the C27-sterol dominance in some animal species have already been known for decades), while on the other hand the work was prominently discussed in Science along with the Michellod et al. study, highlighting its potential importance. I therefore think that Nature Communications is a suitable outlet for the study, but suggest to tone down some of the inferences regarding paleo-ecological implications.

When considering the conclusions and implications made by Brunoir and colleagues, one should look most critically at the main inferences: 1) C27 sterols cannot be considered a shared trait of Neoproterozoic eumetazoans, 2) C28+ steranes do not explicitly indicate the presence of fungi or green algae, 3) above refutes the underlying hypothesis that drives the interpretive paradigm of Neoproterozoic biomarker interpretations and that this 4) rewrites the molecular fossil record:

1) In my opinion, the first inference that not all Neoproterozoic eumetazoans should be expected to necessarily have produced C27 sterols is largely correct and very important. Although this has already been highlighted by Michellod et al. (2023) and a number of extraction-based studies that reported the occurrence of C28+ sterols in various animals, it is worth re-iterating this point through these independent genetic analyses.

2) The inference that C28+ sterols not explicitly indicate fungal or algal sources is certainly correct, but trivial and certainly not a new discovery of the study. For example, Kodner et al. (2008) showed that C28+ steranes are also produced by algae other than Chlorophyta. It is long-known that sponges produce diverse sterols (including C28 and C29) and the senior authors of the study (e.g. in Love et al. 2009, and Gold et al. 2016) prominently argued for important demosponge contributions to the Neoproterozoic sterol-pool, and the close yet unicellular animal relatives Choanoflagellates and fungi are known to produce C28 steranes (Kodner et al. 2008b), while Nettersheim et al. (2019) showed that C28+ sterols are widely distributed across the phylogenetic tree of Rhizaria that they interpreted as potentially important contributors to the Neoproterozoic steroid inventory. This list could be extended, so that it should be clear that C28+ steranes do not explicitly indicate the presence of fungi or green algae in the first place and there should be very few workers that would make such a claim. Also, (at least in the main text) I did not find any mentioning of fungi in refs 4 and 6, so I am not sure where the fungi idea that seems to be central to the main conclusions is coming from (e.g. with fungi and green algae expanding around 663 Ma 4,6). Sure, most fungi produce C28 sterols so that these are often associated with potential fungal sources, but I am not aware of claims that Neoproterozoic C28 steranes can be attributed to fungi (at least with any more confidence than to potential algal or heterotrophic protist sources). If this aspect is important, it would be good to clarify if or when C28 steranes are used as a fungi-marker.

3) I am therefore a little lost as to what underlying hypothesis for Neoproterozoic biomarker interpretations Brunoir and colleagues refer to that could be refuted by their new discovery. Since it is not directly stated, I assume that the underlying hypothesis meant is that fossil C27 sterols are typically attributed to red algae and possible animal sources and fossil C28 sterols from fungi and C29 sterols from green algae. I cannot see how the discovery of additional potential sources could change such interpretations. It is widely known and considered in most paleo-ecological interpretations that there are a variety of potential sources for each steroid-type (and almost any other biomarker), so the most plausible ecological interpretations are carefully drawn based on detailed geological/evolutionary context and accompanying data (e.g. trends in microfossil record). For example, C29 sterols cannot only be sourced from green algae, but also land plants and a variety of heterotrophic protists and even some animals (such as sponges) and it is not possible to determine from the presence of fossil C29 sterols alone which of these (or other) groups of organisms contributed to the Neoproterozoic C29 sterol pool (probably multiple groups). Yet, considering that primary producers such as algae often

contributed a large proportion of the exported carbon and in view of fossil and molecular clock evidence for (pre) Neoproterozoic origins of marine green algae while prominent land plant contributions seem unlikely until the Devonian (Schwark and Empt, 2006), the C29 sterol dominance in Ediacaran to early Phanerozoic marine sediments is most plausibly explained by a dominance of green algae among the eukaryotic primary producers. Other groups may also have contributed, but without other lines of evidence we cannot state with any confidence that there actually were substantial contributions from any group other than green (or possibly some other) algae. With the new data from the current manuscript and that of Michellod et al. (2023), it now becomes plausible that early eumetazoans (ie animals other than sponges) might also have contributed to the Neoproterozoic C28+ steroid pool. But since they may (probably more likely still) have also contributed to the C27 steroid pool (that reaches even deeper into the Neoproterozoic), I cannot really see any fundamental implications for paleo-ecological interpretations. And C28+ sterols are long-known from some animals (even if these were enriched by dietary uptake or symbiont metabolites part of the ancient C28+ pool could accordingly still be derived from early animals; see review by Goad 1981), but as long as we cannot distinguish animal from algal signatures, any such potential animal contributions remain speculative if not accompanied by other lines of evidence. I strongly recommend to clarify what exactly is meant by the underlying hypothesis are that are supposedly refuted by the new discovery and why/how the hypothesis can now be refuted (if that really is the case).

4) Therefore, I also cannot see in which way the discovery could substantially rewrite the molecular fossil record and strongly suggest to change the title as I find such a strong inference very inappropriate in above descript context.

The only hypothesis I can see that may be challenged is that of C27 sterols being a shared trait of Neoproterozoic eumetazoans, but it is unclear to me for which interpretations this hypothesis is fundamental. I assume the authors allude to the interpretations of the strong C27 sterol-dominance of the Ediacaran fossil Dickinsonia that was used by Bobrovskiy et al. (2018) to infer an animal-affinity of the fossils based on the typical C27 sterol dominance in most animals and differential sterol-patterns in other groups that might have a similarly deep evolutionary history and conceivably have produced fossils with similar morphology such as lichen or certain protozoa such as Foraminifera. But the (not really new) realization that not all extant animals have a (near-exclusive) C27 sterol dominance and that accordingly not all early animals necessarily had exclusive C27 sterol dominance, does in my view not mean that early animals would have been unlikely to produce similar (typically C27 dominated) sterol signatures as most of their modern descendants. It just allows for Neoproterozoic C28+ sterols to potentially also derive from eumetazoans (depending on the validity of early origin and vertical inheritance of eumetazoan SMT that contrasts in the current study and the one of Michellod et al. (2023) that instead argue for horizontal gene transfer rather than vertical inheritance).

In the special case where recently fossil sterol signatures in Ediacaran fossils were used to infer the likely feeding strategies and potential gut-content of early animals (Bobrovskiy et al. 2022), one could argue that the new smt discoveries highlight the potential for C28 sterols in the fossil representing animal rather than dietary lipids. However, considering that not only steroid homolog but also isomerization patterns were employed for the interpretations of Bobrovskiy et al. (2022), it is not obvious if or how much this would change ecological interpretations—certainly not enough to argue for a more general rewriting of the molecular fossil record. It should also be noted that it was recently inferred that even some bacteria may be potential sources of C28+ steranes (Brown et al. 2023), so the realization that not only various types of algae and protozoans, sponges and bacteria but maybe also eumetazoans are among the potential sources of Neoproterozoic C28+ steranes does not really change the plausibility argument that primary producers (alage) are most likely the dominant sources of the sedimentary signatures unless there is independent evidence for alternative sources. As discussed above, additional input from other groups of organisms is possible, but currently cannot be further constraints. Therefore, I strongly suggest to revise the implications for the interpretation of the molecular fossil record.

It may also be worth mentioning the recent discovery of vary promiscuous bacterial SMTs in the discussion of smt gene copy number and side-chain alkylation even if the authors were careful to only talk about eukaryotic smts. In my view, the promiscuous nature of SMTs in both demosponges and bacteria (plus probably promiscuous plant enzymes such as the biofunctional SMT2 in *A. thaliana* mentioned later in the manuscript and the annelid inferences by Michellod et al.) puts some doubt on interpretations of sterol biosynthetic capabilities and smt copies that needs to be resolved in future studies. This may also be important for the broad overlap of the putative animal SMT duplication and the Neoproterozoic rise of C28 and C29 steranes as even in the current study (not considering the difficulties of time-calibration for molecular clocks), the animal SMT1 is suggested to have evolved sometime in the Meso- or Neoproterozoic in which case eumetazoan production of C28+ sterols might greatly precede the Neoproterozoic C28+ sterane record.

Regarding the phylogenetic interpretations, it is important to note that Michellod et al. interpret consistent incongruencies throughout most of the C24-SMT tree to be difficult to reconcile with direct inheritance from a common eukaryotic ancestor. Instead, they consider repeated, independent events of lateral gene transfer as the more likely explanation. It is good to see that Brunoir et al. conduct additional reconciliation analyses to help distinguish between vertical inheritance and horizontal gene transfer. However, as I understand it is the purpose of these analyses to reconcile the gene with the species tree (i.e. it is the underlying assumption that nodes with low statistical support a erroneous, but I assume this is a hypothesis that would have to be tested with higher quality data), so in my understanding it should not be surprising to have the reconciled SMTs form a single clade. This could be closer to the actual evolutionary smt history than the untreated data, but it could as well create a false impression (in particular of the contrasting interpretation in Michellod et al. for the broader phylogenetic data set). I cannot assess the reliability of these analyses but considering the degree of data cleaning conducted in the present study with the exclusive focus on animals rather than taking into account the likelihood of horizontal SMT transfer indicated by the wider phylogenetic dataset of Michellod et al. (2023), I think that the authors might consider to be a little more cautious with the inferences from the reconciled tree or add some other independent analyses that verify the inferences.

At the moment, divergence from the typical C27-dominance as a hallmark of most animals may be a special case that potentially requires particularly circumstances (e.g. unusual feeding modes or interaction with symbionts), and to me it is unclear how likely it is that early animals even had the capacity to produce C28+ sterols (in view of Michellods contrasting interpretation) or if/under which (environmental) conditions they might have done so in substantial quantities. C27 sterol dominance may still have been a shared trait of (at least most) Neoproterozoic eumetazoans (in particular based on the results of biomarker analyses on Ediacara Biota fossils)—the current study just adds some more uncertainty to this inference. To me it still seems likely that a large proportion of early animals (albite possibly not all of them) produced similar sterol signatures as most of their extant relatives. Thus at the moment I don't see any basis for rewriting the molecular fossil record, but the new studies highlight the need for additional studies to constrain the evolutionary patterns of sterol biosynthesis in animals and other eukaryotes and even bacteria (that may play an important role in many animals with unusual sterol patterns such as sponges). In my view, biomarkers have to be carefully interpreted in sedimentological and, in case of Ediacara Biota, morphological context: as there is no C27 sterol animal biomarker as such, the capacity of potential C28+ production in early animals (if correctly interpreted) would neither force a reconsideration of what counts as an animal biomarker, nor a reinterpretation of the Neoproterozoic sterane fossil record.

In the discussion, I would similarly advise to be more careful with the ecological implications: for example I would think that it is fair to say that the study indicates that early animals may have had the genes to synthesize C28+ sterols, but I think with the current data this question is not yet satisfactorily resolved. I am not sure where in the literature it is stated that cholesterol biosynthesis is the ancestral condition of the Eumetazoa, again it would be good to clarify this important aspect. As discussed above, I would still think its likely that a lot of early animals had similar sterol signatures as

many of their modern relatives (ie C27 dominance may already have been common). With the new discovery it is now more likely that some also produced substantial amount of C28+ sterols, but currently there is no support for this from the fossil record with the known examples of probable Ediacaran animals seemingly producing similar (C27 dominated) signatures as most of their modern relatives (Bobrovskiy et al. 2018; 2023). One can speculate that availability of dietary sterols might favour multiple smt loses as proposed by Brunoir and colleagues, but it seems (at least) equally likely that C27 dominance is so widespread across the eumetazoans because it is either an ancestral feature or often advantageous (and may have been so for many early relatives that then also mostly produced C29 sterols even if they already had the genetic capacity to produce C28+ also). Modern eumetazoans can turn to abundant phytosterols in their diets/environments, yet the large majority prefers cholesterol biosynthesis and/or conversion of dietary sterols to cholesteroloids (Martin-Creuzburg and Elert, 2009). I find these patterns difficult to reconcile with evolutionary adaptations to abundant C28+ biosynthesis: if their ancestors generally invested a lot of energy and resources to specifically alkylate the sterol side-chain, why do most of their descendants invest similar energy and resources to de-alkylate the side-chain of dietary sterols?

In view of C28+ sterol production or even widespread enrichment (from dietary sources) still appearing to be a rather special case in eumetazoans and absence of evidence for C28+ production in likely Neoproterozoic animal fossils, I think it is too early to infer vastly different sterol signatures in early animals compared to the large majority of their modern relatives. I think dissimilar sterol patterns of early eumetazoans are an interesting hypothesis suitable for publication in Nature Communications and one that should be considered in future biomarker interpretations and in particular follow-up studies that further constrain the evolutionary relationship of sterol production in animals and other groups of eukaryotes and bacteria. But I think it should be presented as a hypothesis arising from the new analyses. Even if correct, I cannot really see how substantial C28+ production in many early animals would rewrite the molecular fossil record and therefore suggest to instead rewrite the title and those parts of the manuscript highlighted above where the impression is given that the presence of C27 sterols (alone) is currently used as an animal biomarker or that similarly C28 and C29 sterols at would require (exclusive) fungal or green algal sources when multiple potential sources are long-known for virtually all sterol-types and routinely assessed in plausibility arguments that take into account much more than just presence or absence of a particular sterol (lipid) type. For example, in the last paragraph I would suggest replacing 'clearly demonstrate' with a less drastic wording such as 'indicate' and rephrase the following sentence: 'This puts our current interpretation of the early eukaryotic biomarker record' in a way that it more specifically states where current interpretations may be questioned (i.e. expected sterol contributions of early animals rather than the early eukaryotic biomarker record in general). But with such amendments, I think the manuscript is suitable for publication in Nature Communications.

Review References

- 1 Kodner, R. B., Pearson, A., Summons, R. E. & Knoll, A. H. Sterols in red and green algae: quantification, phylogeny, and relevance for the interpretation of geologic steranes. *Geobiology* 6, 411-420, doi:10.1111/j.1472-4669.2008.00167.x (2008).
- 2 Kodner, R. B., Summons, R. E., Pearson, A., King, N. & Knoll, A. H. Sterols in a unicellular relative of the metazoans. *Proceedings of the National Academy of Sciences* 105, 9897-9902 (2008).
- 3 Love, G. D. et al. Fossil steroids record the appearance of Demospongiae during the Cryogenian period. *Nature* 457, 718-721 (2009).
- 4 Gold, D. A. et al. Sterol and genomic analyses validate the sponge biomarker hypothesis. *Proceedings of the National Academy of Sciences*, doi:10.1073/pnas.1512614113 (2016).
- 5 Nettersheim, B. J. et al. Putative sponge biomarkers in unicellular Rhizaria question an early rise of animals. *Nature Ecology & Evolution* 3, 577-581, doi:10.1038/s41559-019-0806-5 (2019).
- 6 Bobrovskiy, I. et al. Ancient steroids establish the Ediacaran fossil Dickinsonia as one of the earliest animals. *Science* 361, 1246-1249 (2018).
- 7 Brocks, J. & Bobrovskiy, I. Some animals make plant sterols. *Science* 380, 455-456, doi:10.1126/science.adh8097 (2023).

- 8 Michellod, D. et al. De novo phytosterol synthesis in animals. *Science* 380, 520-526, doi:doi:10.1126/science.add7830 (2023).
- 9 Bobrovskiy, I., Nagovitsyn, A., Hope, J. M., Luzhnaya, E. & Brocks, J. J. Guts, gut contents, and feeding strategies of Ediacaran animals. *Current Biology*, doi:https://doi.org/10.1016/j.cub.2022.10.051 (2022).
- 10 Brown, M. O., Olagunju, B. O., Giner, J.-L. & Welander, P. V. Sterol methyltransferases in uncultured bacteria complicate eukaryotic biomarker interpretations. *Nature Communications* 14, 1859, doi:10.1038/s41467-023-37552-3 (2023).
- 11 Martin-Creuzburg, D. & Elert, E. v. Ecological significance of sterols in aquatic food webs. (Springer, 2009).
- 12 Schwark, L. & Empt, P. Sterane biomarkers as indicators of palaeozoic algal evolution and extinction events. *Palaeogeography, Palaeoclimatology, Palaeoecology* 240, 225-236, doi:http://dx.doi.org/10.1016/j.palaeo.2006.03.050 (2006).
- 13 Goad, L. Sterol biosynthesis and metabolism in marine invertebrates. *Pure and Applied Chemistry* 53, 837-852 (1981).

Reviewer #4:

Remarks to the Author:

This article presents a nice complement to a study recently published by Michellod et al about the rôle of SMT genes in annelid sterol methylation. More specifically, it provides a first biochemical characterization of an annelid SMT expressed in an eukaryotic heterologous system, detailed genomic information and extensive phylogenetic analysis on some sequences and first glance at potential 3D structures of those proteins.

There is a major point to be revised concerning the phylogenetic analysis, and several other minor issues to so

Major point

Page 8 : « all animal SMTs formed a single clade (Figure S2) »

This is not really what the three shows. The yellow clade first divides into two groups, one with the rotifera sequences, and an other containing not only sponges, corals and annelids, but also fungi and various lineages of photosynthetic eukaryotes.

And as the authors acknowledge themselves later in the paragraph, even in this second group, the animal sequences are not fully consistent with the species phylogeny. But once again the text does not completely reflect what the three shows :

« Instead we find statistical support for two clades: one containing sea sponges and corals, and the other containing sea sponges, rotifers and annelids. »

What I see is one clade indeed containing sponges and corals, with a weak branch support (68), and another one more solid (branch support 94) containing only annelids sequences. Withing the sponge and coral group, sponges are splitted into four clades. Given the species distribution in sponges, and expected variation in sequence data in species that are expected to be heterozygotic, I see no evidence for « a gene duplication event occurring before sea sponges diverged from the other living animals. ».

Actually, looking at Figure 5, I understood that the authors used sequences highlighted in yellow in Figure S2 to build the final sequence dataset for the three shown in Figure 5. However, it is not possible to assess the solidity of this final tree because no branch support values are shown. Instead of this, only time divergence estimates are shown. Therefore, no convincing evidence is provided about the claim of an early SMT duplication during animal evolution. A closer look at the tree indicates, that the sponge SMT sequences grouping with corals, those from *Neopetrosia* and the *Haliciona* + *Amphimedon* clades, are the ones that were already attracted to cnidarian in Figure S2. And there belong to different genres than sequences interpreted as « Animal SMT2 ». All this being said, the

most reasonable interpretation for me would be that it's difficult to resolve sponge monophyly because of the sum of variation in sequence evolution rates, errors in transcriptomic data etc.... Regarding the outgroups, please also note that the shown divergence times seem to be wrong, because a SMT duplication seem to have occurred within the diaphoretickes lineage before the divergence between archeplastids and SAR. According to timetree.org, the median estimated divergence time between Arabidopsis and Aureococcus is 1276 MYA.

Also, although I appreciate that this could turn on to be complicated, it should be worth to comment about the animal SMT clades found here and those reported in the Michellod study, were the contamination issue was also openly discussed. In my impression, the authors found a quite clear procedure to eliminate some contaminant sequences, and it would be worth to clearly illustrate how it can be useful, and which previously reported animal clades may be artefactual.

Minor points

Abstract :

segmented worms, an advanced group of "higher" animals (clade Eumetazoa)
→ annelids, also called segmented worms, that belong the Eumetazoa clade
page 2 : animals and red algae → animals and multicellular red algae (see Kodner et al., Geobiology 2008, doi : 10.1111/j.1472-4669.2008.00167.x for reports of methylated sterols in unicellular red algae)

Page 2 :

C28, sterols are the dominant sterols found in fungi, and C29 sterols are common in green algae and plants

→ C28, sterols are the dominant sterols found in most fungi, and C29 sterols are common in green algae, plants, and some specific fungal lineages (glomeromycota, pucciniomycotina, monoblepharidales). See Weete et al., Plos One 2010 (doi:10.1371/journal.pone.0010899) for details on fungi.

Figure 1 : Barney Creek Steroids are not mentioned at all in the text. Now that the Brocks et al paper is out in Nature, it should be cited (doi :10.1038/s41586-023-06170-w) and discussed.

« For example, there are many living groups of eukaryotes besides green algae and plants that produce C29 sterols—including certain fungi, choanoflagellates, diatoms, and ichthyosporeans »
Brown algae could be added to this list of taxa. Moreover, regarding the use of paleomarkers, sterols from diatom or brown algal origin should not be a problem for interpreting neoproterozoic samples, since they are believed to arise only much later, during the mesozoic.

publicly available genetic data → genomic data
are part of the higher animals (clade Eumetazoa) → are part of the Eumetazoa clade
genetic databases → sequence (or genomic) databases (several occurrences)

Figure 2 : To improve readability I suggest aligning the methyltransferase and C-term domains. The very long N-term of *Paraescarpa echinospica* could be abbreviated like this :

-----//-----

One obvious case of incomplete protein after such an alignment is the second sequence form *Eisena fetida*, which could not be expected to carry out the canonical catalytic activity if lacking half of its methyltransferase domain. So maybe it should be discarded, as other unshown partial sequences.

Figure 3 (E-F) →(E-G, I-K)

alignment of annelid proteins → alignment of the three above annelid proteins

Page 5 : « This supports recent results by Michellod et al. »

→ reference should be updated with the published article in Science.

Given that the authors present inferences from modelling and not experimental results, I would tune down the phrasing to be a little bit more careful about the conclusions :

« The *C. teleta* SMT and one of the two *P. dumerilii* proteins are more structurally similar to the bifunctional SMT2 than ERG6, as demonstrated by their lower RMSD scores, and is particularly noticeable in the alpha helices in the C-terminal domain. »

→ The proteins may be more similar to..., as suggested by...

Similarly, on page 6 : After characterizing the *C. teleta* SMT → After inferring the structural characteristics of the *C. teleta* SMT

Page 7 :

demonstrating that multiple annelid species can synthesize complex sterols de novo

→ demonstrating that multiple annelid species can methylate sterols

Figure 4D : I suggest putting the original spectra of analytic standard vs sample fragmentation spectra in supplementary data, in order to let the reader judge on its own about how much both spectra are comparable.

Typos

Figure S4 legend : reproduced → reproduced

Page 6 : ERG6- yeast are viable → ERG6- yeasts

POINT-BY-POINT RESPONSE TO THE REVIEWERS' COMMENTS

GENERAL REPLY TO REVIEWERS: CHANGE IN PAPER EMPHASIS

As part of our revision process, we have performed a major rewrite of the abstract, introduction, and conclusion. While our analyses and major results have not changed, we have shifted the focus to better dialoged with the recently published results of Michellod *et al* (2023). We now emphasize where our results disagree, particularly on the issue of horizontal versus lateral gene inheritance of animal *smt* genes. We provide strong support for the common ancestry of animal *smt* genes. As a result, we no longer argue that our results necessitate a “rewriting” of the molecular fossil record. We think this change in focus obviates many of the reviewer concerns, which we address point by point below.

REVIEWER COMMENTS (reproduced in blue)

Reviewer #1 (Remarks to the Author):

Brunoir et al address a key issue in paleontology: how to associate molecular fossil biomarkers to specific phylogenetic groups. This is especially critical for tracking first eukaryotes and particularly first animals along the Neoproterozoic-age (~1,000-541 Ma), where animal fossils are extremely scarce. By means of phylogenomics and molecular clock studies with a gene encoding a sterol methyltransferase (*smt*), they suggest a gene duplication event in the earliest animals that overlaps with a prominence of C28 and C29 steranes in the fossil record ~663-635 Ma (coincident with the onset of Ediacaran period), so far attributed to fungi (C28) and green algae (C29). The results bring the authors to reject the so far accepted hypothesis that cholesterol (C27) biosynthesis is the ancestral condition of the Eumetazoa, and suggest some animal lineages as potential relevant source of C28 and C29 steranes.

This reviewer finds this work highly relevant for the study of ancient life and considers it sheds light to associate lipid biomarkers to their biological producers. Also, the work shows how phylogenomics and biogeochemistry complement each other to decipher the record of life on Earth.

We thank this reviewer for the positive feedback.

Minor comments:

Introduction

The work by Michellod et al. (ref. 22) should be mentioned here, to let the reader have more data about the *smt* gene and its presence in annelids. (Note: this reviewer has nothing to do with that work nor the group.)

The introduction has been completely rewritten to dialogue with Michellod et al. Please see our opening paragraph in this document for more details.

Discussion

Where the authors say:

“We therefore reject the hypothesis that cholesterol biosynthesis is the ancestral condition of the Eumetazoa”, this reviewer misses additional comments on cholesterol in the new scenario depicted.

This sentence has been rewritten. Instead, we emphasize that the earliest eumetazoans had at least one *smt* gene and were capable of synthesizing higher sterols. We no longer make a claim about the dominant sterols in early eumetazoans.

Fig. 3 legend: where it says (E-F), it should be (E-G); and where it says (I-J), it should say (I-K).

Correction made. Thank you for catching this error.

Fig. 4 legend: where it says: (A) Partial total ion chromatogram, should not say (A) Partial total gas chromatogram?

Correction made.

Reviewer #2 (Remarks to the Author):

In their manuscript: An ancient gene duplication in animals rewrites the molecular fossil record, Brunoir and colleagues report the discovery of the gene 24-C sterol methyltransferase (smt) required for phytosterol production in certain animals (segmented worms) and use a molecular clock estimation to conclude that the animal-specific smt derived from a Neoproterozoic gene duplication event that overlaps with a rise in the abundance of C28 and C29 steranes observed in the rock record. They use these inferences to conclude that C27 sterol-production should not be viewed as a shared trait of Neoproterozoic eumetazoans, and sterane with 28 or more carbon atoms are not necessarily derived from fungi or green algae. While admitting that these results do not necessarily contradict the emerging picture of Neoproterozoic life informed by molecular fossils, Brunoir and colleagues conclude that their findings refute the underlying hypothesis that drives the interpretive paradigm, going as far as prominently claiming a rewriting of the molecular fossil record in their manuscript title.

This reviewer will be pleased to know we no longer argue for “rewriting” the molecular fossil record. Please see our introductory paragraph in this document for more details.

Considering these potentially very far-reaching inferences, I was asked to assess the implications of the new discovery for interpretations of the molecular fossil record, and will not assess the underlying genetic analyses in detail. When considering the potential relevance of the study of Brunoir and colleagues and the suitability for publication in *Nature Communications*, it should be noted that a very similar study was recently published in *Science* (Michellod et al. 2023) and that the implications of both the Michellod et al. (2023) and the yet unpublished preprint of the here discussed study for interpretations of the molecular fossil record were briefly discussed in the accompanying opinion piece of Brocks and Bobrovskiy (2023). This means, that on the one hand the main discovery, the capacity of phytosterol-production in animals, has already been published (and exceptions to the C27-sterol dominance in some animal species have already been known for decades), while on the other hand the work was prominently discussed in *Science* along with the Michellod et al. study, highlighting its potential importance. I therefore think that *Nature Communications* is a suitable outlet for the study, but suggest to tone down some of the inferences regarding paleo-ecological implications.

We appreciate the reviewer sees *Nature Communications* as an appropriate venue for this study. And as suggested, we have toned down the inferences being made.

When considering the conclusions and implications made by Brunoir and colleagues, one should look most critically at the main inferences: 1) C27 sterols cannot be considered a shared trait of Neoproterozoic eumetazoans, 2) C28+ steranes do not explicitly indicate the presence of fungi or green algae, 3) above refutes the underlying hypothesis that drives the interpretive paradigm of Neoproterozoic biomarker interpretations and that this 4) rewrites the molecular fossil record:

1) In my opinion, the first inference that not all Neoproterozoic eumetazoans should be expected to necessarily have produced C27 sterols is largely correct and very important. Although this has already been highlighted by Michellod et al. (2023) and a number of extraction-based studies that reported the

occurrence of C28+ sterols in various animals, it is worth re-iterating this point through these independent genetic analyses.

We agree, and reiterate this point in our revised conclusion.

2) The inference that C28+ sterols not explicitly indicate fungal or algal sources is certainly correct, but trivial and certainly not a new discovery of the study. For example, Kodner et al. (2008) showed that C28+ steranes are also produced by algae other than Chlorophyta. It is long-known that sponges produce diverse sterols (including C28 and C29) and the senior authors of the study (e.g. in Love et al. 2009, and Gold et al. 2016) prominently argued for important demosponge contributions to the Neoproterozoic sterol-pool, and the close yet unicellular animal relatives Choanoflagellates and fungi are known to produce C28 steranes (Kodner et al. 2008b), while Nettersheim et al. (2019) showed that C28+ sterols are widely distributed across the phylogenetic tree of Rhizaria that they interpreted as potentially important contributors to the Neoproterozoic steroid inventory. This list could be extended, so that it should be clear that C28+ steranes do not explicitly indicate the presence of fungi or green algae in the first place and there should be very few workers that would make such a claim. Also, (at least in the main text) I did not find any mentioning of fungi in refs 4 and 6, so I am not sure where the fungi idea that seems to be central to the main conclusions is coming from (e.g. with fungi and green algae expanding around 663 Ma 4,6). Sure, most fungi produce C28 sterols so that these are often associated with potential fungal sources, but I am not aware of claims that Neoproterozoic C28 steranes can be attributed to fungi (at least with any more confidence than to potential algal or heterotrophic protist sources). If this aspect is important, it would be good to clarify if or when C28 steranes are used as a fungi-marker.

We agree with this critique. We no longer make the claim that C₂₈ sterols are indicative of fungi. Please see the revised introduction and conclusion.

3) I am therefore a little lost as to what underlying hypothesis for Neoproterozoic biomarker interpretations Brunoir and colleagues refer to that could be refuted by their new discovery. Since it is not directly stated, I assume that the underlying hypothesis meant is that fossil C27 sterols are typically attributed to red algae and possible animal sources and fossil C28 sterols from fungi and C29 sterols from green algae. I cannot see how the discovery of additional potential sources could change such interpretations. It is widely known and considered in most paleo-ecological interpretations that there are a variety of potential sources for each steroid-type (and almost any other biomarker), so the most plausible ecological interpretations are carefully drawn based on detailed geological/evolutionary context and accompanying data (e.g. trends in microfossil record). For example, C29 sterols cannot only be sourced from green algae, but also land plants and a variety of heterotrophic protists and even some animals (such as sponges) and it is not possible to determine from the presence of fossil C29 sterols alone which of these (or other) groups of organisms contributed to the Neoproterozoic C29 sterol pool (probably multiple groups). Yet, considering that primary producers such as algae often contributed a large proportion of the exported carbon and in view of fossil and molecular clock evidence for (pre) Neoproterozoic origins of marine green algae while prominent land plant contributions seem unlikely until the Devonian (Schwark and Emt, 2006), the C29 sterol dominance in Ediacaran to early Phanerozoic marine sediments is most plausibly explained by a dominance of green algae among the eukaryotic primary producers. Other groups may also have contributed, but without other lines of evidence we cannot state with any confidence that there actually were substantial contributions from any group other than green (or possibly some other) algae. With the new data from the current manuscript and that of Michellod et al. (2023), it now becomes plausible that early eumetazoans (ie animals other than sponges) might also have contributed to the Neoproterozoic C28+ steroid pool. But since they may (probably more likely still) have also contributed to the C27 steroid pool (that reaches even deeper into the Neoproterozoic), I cannot really see any fundamental implications for paleo-ecological interpretations. And C28+ sterols are long-known from some animals (even if these were enriched by dietary uptake or symbiont metabolites part

of the ancient C28+ pool could accordingly still be derived from early animals; see review by Goad 1981), but as long as we cannot distinguish animal from algal signatures, any such potential animal contributions remain speculative if not accompanied by other lines of evidence. I strongly recommend to clarify what exactly is meant by the underlying hypothesis are that are supposedly refuted by the new discovery and why/how the hypothesis can now be refuted (if that really is the case).

Again, we generally agree with this critique, and we no longer claim to refute any “underlying hypothesis” about the biomarker record.

4) Therefore, I also cannot see in which way the discovery could substantially rewrite the molecular fossil record and strongly suggest to change the title as I find such a strong inference very inappropriate in above descript context. The only hypothesis I can see that may be challenged is that of C27 sterols being a shared trait of Neoproterozoic eumetazoans, but it is unclear to me for which interpretations this hypothesis is fundamental. I assume the authors allude to the interpretations of the strong C27 steroid-dominance of the Ediacaran fossil Dickinsonia that was used by Bobrovskiy et al. (2018) to infer an animal-affinity of the fossils based on the typical C27 sterol dominance in most animals and differential sterol-patterns in other groups that might have a similarly deep evolutionary history and conceivably have produced fossils with similar morphology such as lichen or certain protozoa such as Foraminifera. But the (not really new) realization that not all extant animals have a (near-exclusive) C27 sterol dominance and that accordingly not all early animals necessarily had exclusive C27 sterol dominance, does in my view not mean that early animals would have been unlikely to produce similar (typically C27 dominated) sterol signatures as most of their modern descendants. It just allows for Neoproterozoic C28+ sterols to potentially also derive from eumetazoans (depending on the validity of early origin and vertical inheritance of eumetazoan SMT that contrasts in the current study and the one of Michellod et al. (2023) that instead argue for horizontal gene transfer rather than vertical inheritance).

We agree that the major contribution from our work deals with horizontal gene transfer versus vertical inheritance. In our revised results, we provide evidence that many genes from Michellod et al. represent contamination from food and/or symbionts, and make the case that our results are more robust. We therefore refute the claim by Michellod *et al.* that eumetazoan animals received their smt gene through horizontal gene transfer. In our revised Discussion, we demonstrate how, based on our molecular clock results, the horizontal gene transfer hypothesis would have no bearing on the Neoproterozoic/Cambrian transition, while our preferred hypothesis does.

In the special case where recently fossil sterol signatures in Ediacaran fossils were used to infer the likely feeding strategies and potential gut-content of early animals (Bobrovskiy et al. 2022), one could argue that the new smt discoveries highlight the potential for C28 sterols in the fossil representing animal rather than dietary lipids. However, considering that not only steroid homolog but also isomerization patterns were employed for the interpretations of Bobrovskiy et al. (2022), it is not obvious if or how much this would change ecological interpretations—certainly not enough to argue for a more general rewriting of the molecular fossil record. It should also be noted that it was recently inferred that even some bacteria may be potential sources of C28+ steranes (Brown et al. 2023), so the realization that not only various types of algae and protozoans, sponges and bacteria but maybe also eumetazoans are among the potential sources of Neoproterozoic C28+ steranes does not really change the plausibility argument that primary producers (algae) are most likely the dominant sources of the sedimentary signatures unless there is independent evidence for alternative sources. As discussed above, additional input from other groups of organisms is possible, but currently cannot be further constraints. Therefore, I strongly suggest to revise the implications for the interpretation of the molecular fossil record.

We allude to Bobrovskiy et al. 2022 in our revised discussion. We do not necessarily agree the isomerization patterns are sufficient to identify dietary sterols, but that is beyond the scope of this paper.

We also cite Brown et al. 2023 as part of a larger point that the number of *smt* genes are insufficient to predict what sterols a prehistoric organism may have produced.

It may also be worth mentioning the recent discovery of vary promiscuous bacterial SMTs in the discussion of *smt* gene copy number and side-chain alkylation even if the authors were careful to only talk about eukaryotic *smts*. In my view, the promiscuous nature of SMTs in both demosponges and bacteria (plus probably promiscuous plant enzymes such as the biofunctional SMT2 in *A. thaliana* mentioned later in the manuscript and the annelid inferences by Michellod et al.) puts some doubt on interpretations of sterol biosynthetic capabilities and *smt* copies that needs to be resolved in future studies. This may also be important for the broad overlap of the putative animal SMT duplication and the Neoproterozoic rise of C28 and C29 steranes as even in the current study (not considering the difficulties of time-calibration for molecular clocks), the animal SMT1 is suggested to have evolved sometime in the Meso- or Neoproterozoic in which case eumetazoan production of C28+ sterols might greatly precede the Neoproterozoic C28+ sterane record.

Agreed. We make a version of this point in the revised discussion.

Regarding the phylogenetic interpretations, it is important to note that Michellod et al. interpret consistent incongruencies throughout most of the C24-SMT tree to be difficult to reconcile with direct inheritance from a common eukaryotic ancestor. Instead, they consider repeated, independent events of lateral gene transfer as the more likely explanation. It is good to see that Brunoir et al. conduct additional reconciliation analyses to help distinguish between vertical inheritance and horizontal gene transfer. However, as I understand it is the purpose of these analyses to reconcile the gene with the species tree (i.e. it is the underlying assumption that nodes with low statistical support a erroneous, but I assume this is a hypothesis that would have to be tested with higher quality data), so in my understanding it should not be surprising to have the reconciled SMTs form a single clade. This could be closer to the actual evolutionary *smt* history than the untreated data, but it could as well create a false impression (in particular of the contrasting interpretation in Michellod et al. for the broader phylogenetic data set). I cannot assess the reliability of these analyses but considering the degree of data cleaning conducted in the present study with the exclusive focus on animals rather than taking into account the likelihood of horizontal SMT transfer indicated by the wider phylogenetic dataset of Michellod et al. (2023), I think that the authors might consider to be a little more cautious with the inferences from the reconciled tree or add some other independent analyses that verify the inferences.

As our revised Results section explains, we find compelling evidence for the common origin of animal SMTs *before* we perform gene tree/species tree reconciliation. We also provide evidence that some of the sequences included in Michellod et al. are not genuine animal sequences. For example, many of the molluscs they identify have publicly available genomes, and querying those genomes shows these species do not have *smt* genes. This strongly suggests the data they collected from “shotgun” sequence projects came from contaminating RNA.

At the moment, divergence from the typical C27-dominance as a hallmark of most animals may be a special case that potentially requires particularly circumstances (e.g. unusual feeding modes or interaction with symbionts), and to me it is unclear how likely it is that early animals even had the capacity to produce C28+ sterols (in view of Michellods contrasting interpretation) or if/under which (environmental) conditions they might have done so in substantial quantities. C27 sterol dominance may still have been a shared trait of (at least most) Neoproterozoic eumetazoans (in particular based on the results of biomarker analyses on Ediacara Biota fossils)—the current study just adds some more uncertainty to this inference. To me it still seems likely that a large proportion of early animals (albite possibly not all of them) produced similar sterol signatures as most of their extant relatives. Thus at the moment I don't see any basis for rewriting the molecular fossil record, but the new studies highlight the need for additional

studies to constrain the evolutionary patterns of sterol biosynthesis in animals and other eukaryotes and even bacteria (that may play an important role in many animals with unusual sterol patterns such as sponges). In my view, biomarkers have to be carefully interpreted in sedimentological and, in case of Ediacara Biota, morphological context: as there is no C27 sterol animal biomarker as such, the capacity of potential C28+ production in early animals (if correctly interpreted) would neither force a reconsideration of what counts as an animal biomarker, nor a reinterpretation of the Neoproterozoic sterane fossil record.

In our revision, we are agnostic about the dominant sterols in early animals. We also now mention that animals may make these higher sterols in very particular developmental or environmental conditions. However, we also note that many eumetazoan lineages have maintained one or more copies of the *smt* gene for hundreds of millions of years. Figuring out when living eumetazoans utilize SMTs will be important for interpreting the fossil record.

In the discussion, I would similarly advise to be more careful with the ecological implications: for example I would think that it is fair to say that the study indicates that early animals may have had the genes to synthesize C28+ sterols, but I think with the current data this question is not yet satisfactorily resolved. I am not sure where in the literature it is stated that cholesterol biosynthesis is the ancestral condition of the Eumetazoa, again it would be good to clarify this important aspect. As discussed above, I would still think its likely that a lot of early animals had similar sterol signatures as many of their modern relatives (ie C27 dominance may already have been common). With the new discovery it is now more likely that some also produced substantial amount of C28+ sterols, but currently there is no support for this from the fossil record with the known examples of probable Ediacaran animals seemingly producing similar (C27 dominated) signatures as most of their modern relatives (Bobrovskiy et al. 2018; 2023). One can speculate that availability of dietary sterols might favour multiple *smt* losses as proposed by Brunoir and colleagues, but it seems (at least) equally likely that C27 dominance is so widespread across the eumetazoans because it is either an ancestral feature or often advantageous (and may have been so for many early relatives that then also mostly produced C29 sterols even if they already had the genetic capacity to produce C28+ also). Modern eumetazoans can turn to abundant phytosterols in their diets/environments, yet the large majority prefers cholesterol biosynthesis and/or conversion of dietary sterols to cholesteroloids (Martin-Creuzburg and Elert, 2009). I find these patterns difficult to reconcile with evolutionary adaptations to abundant C28+ biosynthesis: if their ancestors generally invested a lot of energy and resources to specifically alkylate the sterol side-chain, why do most of their descendants invest similar energy and resources to de-alkylate the side-chain of dietary sterols?

We personally prefer the hypothesis that *smt* genes were lost across many animal lineages around the Neoproterozoic/Cambrian boundary. We can infer from choanoflagellates and sea sponges that ancestral animals were capable of making C₂₈₊ sterols. As addressed in the revised Discussion, the rise of algae in the Neoproterozoic—as evidenced by the rise of C₂₈₊ steranes—suggests a new pool of food for animals to obtain dietary sterols. The diversification of animal feeding strategies in the Neoproterozoic allowed lineages to abandon or simplify their sterol biosynthesis pathway. So in marked contrast to our initial framing, we now argue that our data is quite consistent with our current interpretation of the molecular fossil record!

In view of C28+ sterol production or even widespread enrichment (from dietary sources) still appearing to be a rather special case in eumetazoans and absence of evidence for C28+ production in likely Neoproterozoic animal fossils, I think it is too early to infer vastly different sterol signatures in early animals compared to the large majority of their modern relatives. I think dissimilar sterol patterns of early eumetazoans are an interesting hypothesis suitable for publication in Nature Communications and one that should be considered in future biomarker interpretations and in particular follow-up studies that further constrain the evolutionary relationship of sterol production in animals and other groups of eukaryotes and bacteria. But I think it should be presented as a hypothesis arising from the new analyses.

Even if correct, I cannot really see how substantial C28+ production in many early animals would rewrite the molecular fossil record and therefore suggest to instead rewrite the title and those parts of the manuscript highlighted above where the impression is given that the presence of C27 sterols (alone) is currently used as an animal biomarker or that similarly C28 and C29 sterols at would require (exclusive) fungal or green algal sources when multiple potential sources are long-known for virtually all sterol-types and routinely assessed in plausibility arguments that take into account much more than just presence or absence of a particular sterol (lipid) type. For example, in the last paragraph I would suggest replacing ‘clearly demonstrate’ with a less drastic wording such as ‘indicate’ and rephrase the following sentence: ‘This puts our current interpretation of the early eukaryotic biomarker record’ in a way that it more specifically states where current interpretations may be questioned (i.e. expected sterol contributions of early animals rather than the early eukaryotic biomarker record in general). But with such amendments, I think the manuscript is suitable for publication in Nature Communications.

We hope the revisions, summarized in our previous answers, assuage the reviewer. Please let us know if our new direction raises additional questions or concerns.

Review References

- 1 Kodner, R. B., Pearson, A., Summons, R. E. & Knoll, A. H. Sterols in red and green algae: quantification, phylogeny, and relevance for the interpretation of geologic steranes. *Geobiology* 6, 411-420, doi:10.1111/j.1472-4669.2008.00167.x (2008).
- 2 Kodner, R. B., Summons, R. E., Pearson, A., King, N. & Knoll, A. H. Sterols in a unicellular relative of the metazoans. *Proceedings of the National Academy of Sciences* 105, 9897-9902 (2008).
- 3 Love, G. D. et al. Fossil steroids record the appearance of Demospongiae during the Cryogenian period. *Nature* 457, 718-721 (2009).
- 4 Gold, D. A. et al. Sterol and genomic analyses validate the sponge biomarker hypothesis. *Proceedings of the National Academy of Sciences*, doi:10.1073/pnas.1512614113 (2016).
- 5 Nettersheim, B. J. et al. Putative sponge biomarkers in unicellular Rhizaria question an early rise of animals. *Nature Ecology & Evolution* 3, 577-581, doi:10.1038/s41559-019-0806-5 (2019).
- 6 Bobrovskiy, I. et al. Ancient steroids establish the Ediacaran fossil Dickinsonia as one of the earliest animals. *Science* 361, 1246-1249 (2018).
- 7 Brocks, J. & Bobrovskiy, I. Some animals make plant sterols. *Science* 380, 455-456, doi:10.1126/science.adh8097 (2023).
- 8 Michellod, D. et al. De novo phytosterol synthesis in animals. *Science* 380, 520-526, doi:doi:10.1126/science.add7830 (2023).
- 9 Bobrovskiy, I., Nagovitsyn, A., Hope, J. M., Luzhnaya, E. & Brocks, J. J. Guts, gut contents, and feeding strategies of Ediacaran animals. *Current Biology*, doi:<https://doi.org/10.1016/j.cub.2022.10.051> (2022).
- 10 Brown, M. O., Olagunju, B. O., Giner, J.-L. & Welander, P. V. Sterol methyltransferases in uncultured bacteria complicate eukaryotic biomarker interpretations. *Nature Communications* 14, 1859, doi:10.1038/s41467-023-37552-3 (2023).
- 11 Martin-Creuzburg, D. & Elert, E. v. Ecological significance of sterols in aquatic food webs. (Springer, 2009).
- 12 Schwark, L. & Empt, P. Sterane biomarkers as indicators of palaeozoic algal evolution and extinction events. *Palaeogeography, Palaeoclimatology, Palaeoecology* 240, 225-236, doi:<http://dx.doi.org/10.1016/j.palaeo.2006.03.050> (2006).
- 13 Goad, L. Sterol biosynthesis and metabolism in marine invertebrates. *Pure and Applied Chemistry* 53, 837-852 (1981).

Reviewer #3 (Remarks to the Author):

This article presents a nice complement to a study recently published by Michellod et al about the rôle of

SMT genes in annelid sterol methylation. More specifically, it provides a first biochemical characterization of an annelid SMT expressed in an eukaryotic heterologous system, detailed genomic information and extensive phylogenetic analysis on some sequences and first glance at potential 3D structures of those proteins.

There is a major point to be revised concerning the phylogenetic analysis, and several other minor issues to so

Major point

Page 8 : « all animal SMTs formed a single clade (Figure S2) »

This is not really what the three shows. The yellow clade first divides into two groups, one with the rotifera sequences, and an other containing not only sponges, corals and annelids, but also fungi and various lineages of photosynthetic eukaryotes.

This was a mistake on our part. Figure S2 shows the maximum likelihood tree *prior* to gene tree / species tree reconciliation. In this tree, all animals except rotifers are monophyletic (albeit with weak bootstrap support). We have revised the Results section to properly contextualize Figure S2.

And as the authors acknowledge themselves later in the paragraph, even in this second group, the animal sequences are not fully consistent with the species phylogeny. But once again the text does not completely reflect what the three shows :

« Instead we find statistical support for two clades: one containing sea sponges and corals, and the other containing sea sponges, rotifers and annelids. »

What I see is one clade indeed containing sponges and corals, with a weak branch support (68), and another one more solid (branch support 94) containing only annelids sequences. Within the sponge and coral group, sponges are splitted into four clades. Given the species distribution in sponges, and expected variation in sequence data in species that are expected to be heterozygotic, I see no evidence for « a gene duplication event occurring before sea sponges diverged from the other living animals. ».

Again, this concern comes from an error on our part, conflating the gene tree before and after reconciliation. The above statement is in regard to the tree *after* gene tree / species tree reconciliation. This tree was originally buried in the online data, but is not available in Figure S3. This has been corrected in the revised manuscript and supplement.

The reviewer is correct that high variation and low sampling in sponges could create the false appearance of an early gene duplication. As we now discuss in the revised Results section, the “duplication” is driven by high bootstrap support (=90) uniting Heteroscleromorph sponges (*Neopetrosia*, *Halicliona*, and *Amphimedon*) and cnidarians. In our revision, we note this could be an artifact caused by limited data, and that a fuller analysis of sponges is needed. A duplication is still what our analysis prefers, so we keep that as our current hypothesis. However, as part of our general rewrite (please see our introductory paragraph in this document for details) we no longer emphasize the gene duplication, as it is not relevant to our primary question: whether the eumetazoan *smt* gene was inherited through horizontal gene transfer.

Actually, looking at Figure 5, I understood that the authors used sequences highlighted in yellow in Figure S2 to build the final sequence dataset for the three shown in Figure 5. However, it is not possible to assess the solidity of this final tree because no branch support values are shown. Instead of this, only time divergence estimates are shown. Therefore, no convincing evidence is provided about the claim of an early SMT duplication during animal evolution. A closer look at the tree indicates, that the sponge SMT sequences grouping with corals, those from *Neopetrosia* and the *Halicliona* + *Amphimedon* clades, are the ones that were already attracted to cnidarian in Figure S2. And there belong to different genres than

sequences interpreted as « Animal SMT2 ». All this being said, the most reasonable interpretation for me would be that it's difficult to resolve sponge monophyly because of the sum of variation in sequence evolution rates, errors in transcriptomic data etc....

We think these concerns have been addressed in the revision. Please see our answers to the last two points.

Regarding the outgroups, please also note that the shown divergence times seem to be wrong, because a SMT duplication seem to have occurred within the diaphoretickes lineage before the divergence between archeplastids and SAR. According to timetree.org, the median estimated divergence time between Arabidopsis and Aureococcus is 1276 MYA.

We have revised the molecular clock, placing a new calibration of the ancestor of archeplastids and SAR. It did not impact the major conclusions of our analysis. For good measure, we also redid the analysis using an alternative topology that does not hypothesize an ancestral gene duplication (Figure S6). Again, the results are consistent with our hypothesis that eumetazoan SMTs diversified around the time of higher steranes in the fossil record. This demonstrates the robustness of our results.

Also, although I appreciate that this could turn out to be complicated, it should be worth to comment about the animal SMT clades found here and those reported in the Michellod study, where the contamination issue was also openly discussed. In my impression, the authors found a quite clear procedure to eliminate some contaminant sequences, and it would be worth to clearly illustrate how it can be useful, and which previously reported animal clades may be artefactual.

This has been done in the revised Results section. In it, we provide compelling evidence that some of the sequences included in Michellod et al. are not genuine animal sequences. For example, many of the mollusc they include in their dataset have publicly available genomes, and querying those genomes shows these species do not have SMTs. This strongly suggests the data they collected from “shotgun” sequence projects came from contaminating RNA, and not the animals themselves.

Minor points

Abstract :
segmented worms, an advanced group of “higher” animals (clade Eumetazoa)
→ annelids, also called segmented worms, that belong the Eumetazoa clade

This sentence has been removed in the abstract rewrite.

page 2 : animals and red algae → animals and multicellular red algae (see Kodner et al., Geobiology 2008, doi : 10.1111/j.1472-4669.2008.00167.x for reports of methylated sterols in unicellular red algae)

This change has been made.

Page 2 :

C28, sterols are the dominant sterols found in fungi, and C29 sterols are common in green algae and plants

→ C28, sterols are the dominant sterols found in most fungi, and C29 sterols are common in green algae, plants, and some specific fungal lineages (glomeromycota, pucciniomycotina, monoblepharidales). See Weete et al., Plos One 2010 (doi:10.1371/journal.pone.0010899) for details on fungi.

We feel listing all the fungal clades with dominant C₂₉ sterols is more detail than this sentence warrants. But we have nuanced the sentence to clarify that C₂₈ sterols are dominant in “most” (not all) fungi. We have also added plants to the sentence and added Weete *et al.* to the references.

Figure 1 : Barney Creek Steroids are not mentioned at all in the text. Now that the Brocks et al paper is out in Nature, it should be cited (doi :10.1038/s41586-023-06170-w) and discussed.

Barney Creek steroids and Brocks et al (2023) have been added to the introduction.

« For example, there are many living groups of eukaryotes besides green algae and plants that produce C₂₉ sterols—including certain fungi, choanoflagellates, diatoms, and ichthyosporeans »
Brown algae could be added to this list of taxa. Moreover, regarding the use of paleomarkers, sterols from diatom or brown algal origin should not be a problem for interpreting neoproterozoic samples, since they are believed to arise only much later, during the mesozoic.

Brown algae have been added and diatoms removed from this sentence.

publicly available genetic data → genomic data

This sentence has been removed as part of the rewrite.

are part of the higher animals (clade Eumetazoa) → are part of the Eumetazoa clade

This sentence has been rewritten.

genetic databases → sequence (or genomic) databases (several occurrences)

This change has been made.

Figure 2 : To improve readability I suggest aligning the methyltransferase and C-term domains. The very long N-term of *Paraescarpa echinospica* could be abbreviated like this :

-----//-----

One obvious case of incomplete protein after such an alignment is the second sequence from *Eisena fetida*, which could not be expected to carry out the canonical catalytic activity if lacking half of its methyltransferase domain. So maybe it should be discarded, as other unshown partial sequences.

We have aligned the proteins as advised. We kept the second *Eisena* sequence, as it is included in our phylogenetic analysis, but we have added a “?” to the sequence to clarify that it is incomplete.

Figure 3 (E-F) →(E-G, I-K)

alignment of annelid proteins → alignment of the three above annelid proteins

This change has been made

Page 5 : « This supports recent results by Michellod et al. »

→ reference should be updated with the published article in Science.

This has been done

Given that the authors present inferences from modelling and not experimental results, I would tune down

the phrasing to be a little bit more careful about the conclusions :

« The C. teleta SMT and one of the two P. dumerilii proteins are more structurally similar to the bifunctional SMT2 than ERG6, as demonstrated by their lower RMSD scores, and is particularly noticeable in the alpha helices in the C-terminal domain. »

→ The proteins may be more similar to..., as suggested by....

This change has been made.

Similarly, on page 6 : After characterizing the C. teleta SMT → After inferring the structural characteristics of the C. teleta SMT

This change has been made.

Page 7 :

demonstrating that multiple annelid species can synthesize complex sterols de novo

→ demonstrating that multiple annelid species can methylate sterols

This change has been made.

Figure 4D : I suggest putting the original spectra of analytic standard vs sample fragmentation spectra in supplementary data, in order to let the reader judge on its own about how much both spectra are comparable.

We have added this data as Figure S7.

Typos

Figure S4 legend : reporduced → reproduced

Fixed.

Page 6 : ERG6- yeast are viable → ERG6- yeasts

Fixed.

Reviewers' Comments:

Reviewer #1:

Remarks to the Author:

All my concerns and comments have been addressed.

Thanks

Reviewer #2:

Remarks to the Author:

Congratulations to the revised version: I think the manuscript has improved very substantially! I appreciate that you implemented such substantial changes and recommend publication of the revised manuscript. I am not fully convinced of the vertical SMT inheritance but this will probably be resolved in future studies that I am very much looking forward to. I think for now you make a good case and I like that it now feels like a very balanced discussion. Well done!

Below just a few instances where I noticed that words seem to be missing/typos:

Following our vetting process we were left with smt genes from three clades of eumetazoans—annelids, rotifers and stony corals. Nematode worms also have an SMTlike protein 18, but we did not include it in our analysis because the gene did not cluster with other eukaryote SMTs in our phylogeny, and because the protein is known to catalyze a C-4 methylation step that is distinct from the C-24 methylation seen in genuine 24.

> End of sentence missing a word?

Figure 6 Summary of smt gene loss in the animals. A simplified animal phylogeny, with the hypothesized presence of smt visualized with red lines.

> Blue lines?

SI:

Museum of Evolution, at Uppsala

University, Russia

> Probably in Sweden?

Because red algae are more closely related to than other algae in our dataset, we can use Bangiomorpha for a crown age calibration.

> Missing a word?

We used C.pertoni to calibrate stem group Embryophyta

> typo

Reviewer #3:

Remarks to the Author:

All my comments have been satisfactorily addressed. Given the extensive rewriting of the manuscript, it seems ready for publication now.

RESPONSE TO REVIEWERS' COMMENTS

(Reviewer comments reprinted in blue)

Reviewer #1 (Remarks to the Author):

All my concerns and comments have been addressed.
Thanks

Reviewer #2 (Remarks to the Author):

Congratulations to the revised version: I think the manuscript has improved very substantially! I appreciate that you implemented such substantial changes and recommend publication of the revised manuscript. I am not fully convinced of the vertical SMT inheritance but this will probably be resolved in future studies that I am very much looking forward to. I think for now you make a good case and I like that it now feels like a very balanced discussion. Well done!

Below just a few instances where I noticed that words seem to be missing/typos:

Following our vetting process we were left with smt genes from three clades of eumetazoans—annelids, rotifers and stony corals. Nematode worms also have an SMTlike protein 18, but we did not include it in our analysis because the gene did not cluster with other eukaryote SMTs in our phylogeny, and because the protein is known to catalyze a C-4 methylation step that is distinct from the C-24 methylation seen in genuine 24.

> End of sentence missing a word?

The missing word is “SMT”. This sentence has been fixed.

Figure 6 Summary of smt gene loss in the animals. A simplified animal phylogeny, with the hypothesized presence of smt visualized with red lines.

> Blue lines?

The reviewer correctly notes the color of this figure has changed. This sentence has been fixed.

SI:

Museum of Evolution, at Uppsala

University, Russia

> Probably in Sweden?

Sweden is correct. This sentence has been fixed.

Because red algae are more closely related to than other algae in our dataset, we

can use Bangiomorpha for a crown age calibration.
> Missing a word?

The missing word is “*Bangiomorpha*”. This sentence has been fixed.

We used *C.pertoni* to calibrate stem group Embryophyta
> typo

This sentence has been fixed.

Reviewer #3 (Remarks to the Author):

All my comments have been satisfactorily addressed. Given the extensive rewriting of the manuscript, it seems ready for publication now.